# Psychometric Validation of Senior Perceived Physical Literacy Instrument

**DOI:** 10.3390/ijerph19116726

**Published:** 2022-05-31

**Authors:** Chien-Yu Liu, Linda Li-Chuan Lin, Jiunn-Jye Sheu, Raymond Kim-Wai Sum

**Affiliations:** 1Graduate Institute of Physical Education, Health & Leisure Studies, National Cheng Kung University, Tainan 701, Taiwan; clarissa.liu85@gmail.com; 2School of Population Health, University of Toledo, Toledo, OH 43606, USA; jiunnjye.sheu@utoledo.edu; 3Department of Sports Science and Physical Education, Faculty of Education, The Chinese University of Hong Kong, Hong Kong, China; kwsum@cuhk.edu.hk

**Keywords:** physical literacy, exercise attitude, exercise ability, social skill, active aging

## Abstract

**Aim:** To validate the Senior Perceived Physical Literacy Instrument (SPPLI). **Methods:** In the pilot study, we refined the Perceived Physical Literacy Instrument (PPLI, Cronbach’s α = 0.94, initially designed for adults) into SPPLI using internal reliability, content validity, and construct validity results. A total of 341 older adults recruited from community centers in Southern Taiwan participated in the study. A principle component analysis (PCA) identified three components of SPPLI. **Results:** Eleven items were captured from the 18-item PPLI as the SPPLI (Cronbach’s α = 0.90). SPPLI exhibits three components: attitude toward physical activity, physical activity ability, and sociality around physical activity. Significant differences were discovered in physical activity ability by educational attainment (*p* < 0.05) and in all three components by stage of exercise (maintenance vs. non-maintenance) (*p* < 0.05). The SPPLI possesses proper reliability and validity to assess physical literacy among older adults. **Conclusions:** This instrument is suggested for physical literacy assessments in physical activity programs to assess the needs of older adults and/or the effectiveness of an intervention program that aims to improve the attitude, ability, and sociality of physical activity.

## 1. Introduction

According to *World Population Prospects 2019* [1], sixteen percent of people in the world will be older adults (over 65 years) in 2050, which is almost double the number from 2019. The challenges of aging exist all over the world [2,3]. The World Health Organization (2015) proposed the concept of “healthy aging,” referring to the development and maintenance of functional ability and the improvement of quality of life in older ages. With the ascending trends of an aging society, it is essential to make aging a positive and active process. Therefore, older adults not only need to prevent and control chronic diseases, but must also increase their physical activity [4]. Being able to perform regular physical activity would improve the quality of life of older adults. To realize the enhancement of physical activity, promoting physical literacy is essential [5]. However, no accurate instrument exists for elderly physical literacy assessment.

Physical literacy is “the motivation, confidence, physical competence, knowledge and understanding to value and take responsibility for engagement in physical activities for life” [6]. Physical literacy also refers to the integration of physical, perceptual, knowledge, cognitive, psychological, sociality, and behavioral capabilities, echoing the need for an active, healthy, and fulfilling lifestyle [6,7,8,9,10,11,12,13]. Physical literacy could protect older adults from chronic diseases and disability. Individuals who are physically literate are more likely to engage in lifelong exercise or physical activity, since they believe and value that physical activity could contribute to their health. Across a life span, physical literacy is more critical to the elderly than people of other ages [14].

To practically assess physical literacy, a valid and reliable instrument is essential. It is also critical that such a measurement of physical literacy is designed to address the stages in life. For children aged 8–12 years, the Canadian Assessment of Physical Literacy—Second Edition (CAPL-2), including a self-administered questionnaire, a compound fitness test, a knowledge test, and a daily activity log to reflect children’s physical literacy, is available to assess the four dimensions of children’s physical literacy, i.e., motivation and confidence, physical competence, knowledge and understanding, and daily behavior [15]. As for adults, the Perceived Physical Literacy Instrument (PPLI) is an instrument employed to assess an adult’s physical literacy in three dimensions, i.e., sense of self and self-confidence, self-expression and communication with others, and knowledge and understanding [16]. However, there is not yet a measurement instrument for the physical literacy of seniors.

Jones [14] reviewed seven studies and developed a physical literacy model for older adults that included motivation and confidence, physical competence, knowledge and understanding, and personality for engagement in physical activity for life as the four core elements. This model matched the three dimensions validated in the PPLI for adults, while there is still a need to design a detailed questionnaire for verification. For older adults, physical literacy is more about the integration of different capabilities to improve one’s lifestyle with age, continue to interact with the environment, and engage in physical activity for life [13,14]. We hypothesized that physical literacy in adults and seniors has different core factors and, thus, invites the need to define a Senior Perceived Physical Literacy Instrument (SPPLI) that considers motivation, confidence, knowledge, physical behavior, and affective and cognitive elements. Therefore, this study aimed to validate the SPPLI and distinguish the key elements of the physical literacy of seniors. The researchers assume that the SPPLI is an effective instrument to assess the physical literacy of older adults.

## 2. Methods and Materials

The protocol for this study involves a non-experimental cross-sectional survey. A descriptive analysis of SPPLI total and component scores was conducted to show the characteristics of participants by age, gender, education level, and exercise stages.

### 2.1. Research Materials

With the authors’ permission, the PPLI (Cronbach’s α = 0.94) developed by Sum [16] was utilized as the basis of the SPPLI. The PPLI was developed through a comprehensive literature review, focus group interviews, content analysis, clarification of concept attributes, theme formation, and a pilot test to evaluate item clarity, receive feedback, and establish the 5-point Likert scale with anchors from “strongly disagree” to “strongly agree.” A four-member panel of sports science, physical education, health education, and instrument development experts with published peer-refereed journal articles evaluated the instrument, revised after the pilot study for item wording, instrument length, clarity of the statements, and response format. An 18-item initial instrument was then ready for validation. After reliability and validity testing, the final version of PPLI contained nine items covering three dimensions of physical literacy: sense of self and self-confidence, self-expression and communication with others, and knowledge and understanding. Due to the scientific merits in the development of PPLI, we adopted this 18-item initial instrument as the basis of SPPLI validation.

### 2.2. Procedures

Two stages of participant recruitment, including the pilot study and final validation, were executed to ensure the quality of the SPPLI. Participants who did not have apparent cognitive decline were invited to review and complete the informed consent before they responded to the PPLI or SPPLI. The trained research staff distributed and collected the questionnaires. They also read the questions out loud to the participants who were not literate enough to understand the questions. The members of the research team have certain background knowledge on health promotion issues for the elderly, and this team included six graduate students in the fields of public health and sports health promotion. Before the experiment, the research team were trained to strengthen the members’ cognition of the physical literacy of the elderly, improve the questioning methods of each member, and consolidate their understanding of the instrument questions. The study was reviewed and approved by the Human Experiment and Ethics Committee of National Cheng Kung University Hospital (No. A-ER-109-559) prior to the recruitment of participants.

### 2.3. Participants

The pilot study included 238 older adults above 65 years old who were recruited from community centers in Southern Taiwan to complete the PPLI. Cronbach alpha and exploratory factor analysis were conducted using the data from the pilot study. To develop the SPPLI from the PPLI, the content validity was examined by a panel of four experts in the physical education and health behavior fields, in which they assessed the breadth and appropriateness of the instrument. Principle component analyses (PCA) were conducted to assess construct validity.

After the pilot study resulted in the 11-item SPPLI, an additional 103 older adults completed the SPPLI. The condensed 11 items in the SPPLI used exactly the same wording and response options as in the pilot study. Therefore, the responses of 238 participants in the pilot testing were merged with the additional 103 participants for analysis. SPPLI validation was, therefore, conducted using the data of 341 participants.

### 2.4. Data Analysis

To test the psychometric properties of the SPPLI, SPSS 20.0 was used to conduct the statistical analysis. The type I error was set at 0.05. Descriptive statistics including frequency, proportion, mean, and standard deviation were calculated to describe the score distribution.

An exploratory factor analysis (EFA) was conducted to assess the construct validity, and Cronbach’s alpha was calculated to assess the internal consistency reliability in both the pilot study using the PPLI and the final validation of the SPPLI. A Cronbach’s alpha value higher than 0.70 would be considered acceptable, reflecting the internal correlation among questions of the instrument [17].

One-way ANOVA and Student’s t-test were used to analyze the difference in SPPLI total and component scores by sex, education level, and exercise maintenance stage.

## 3. Results

A convenience sample of 341 older adults, including 264 women (78%) and 77 men (22%) (aged 65–95), participated in this study (Table 1). No gender difference in SPPLI total score and its three component scores was discovered. The mean age of the participants was 74 (SD = 7.5 years). Most participants had completed primary education, and 75% of them had also graduated from junior high schools and were moderately literate. About three quarters (72%) of participants exercised regularly for six or more months (i.e., in the maintenance stage, according to the Stage of Change Model by [18]

To develop the SPPLI, the researchers conducted internal consistency reliability and content and construct validity assessments. In the pilot study, an initial PCA was conducted to assess the PPLI’s suitability in older adults. As a result, PCA identified PPLI items #12 and #13 as unsuitable and, thus, they were removed (Table 2). After this initial removal, the 16 items left were validated again by PCA. The results led to the removal of the items #4, #5, #7, #8 and #18 because of the factor loading being lower than 0.7 in consideration of the internal consistency reliability and expert panel recommendations (Table 3).

After the two removals, the remaining 11 items resulted in the SPPLI. The Cronbach’s alpha for this finalized SPPLI was 0.90. The items’ mean scores ranged from 3.8 to 4.5. Including the additional participants, the construct validity of the finalized SPPLI was assessed by PCA. Three components (Table 4), with eigenvalues greater than 1.0, account for 72.5% of the total variance. Factor loading of the scale ranged from 0.72 to 0.85. Cronbach’s alpha of internal consistency ranged from 0.80 to 0.90, meeting the criterion of 0.70 [17]. The SPPLI has 11 questions in three dimensions, which were named “attitude toward physical activity,” “physical activity ability,” and “sociality around physical activity,” in consideration of previous physical literacy research [8].

The first component includes items #1, #2, #3, #4, and #5, which are “I can turn doing sports into an on-going habit of life,” “I have a mindset for lifelong sports,” “I am willing to do sports for better health,” “I am aware of the benefits of sports related to health,” and “I establish friendship through sports,” respectively (Table 5). Component #1 expounds older adults’ attitudes toward physical activity or exercise with the score ranging from 5 to 25 (mean = 21.6, SD = 2.9). The second component includes items #6, #7, #8, and #9, which state “I possess adequate fundamental movement skills,” “I am physically fit, in accordance to my age,” “I am able to apply exercise knowledge in the long run,” and “I am able to apply learnt motor skills to other physical activities,” respectively. Component #2 illustrates older adults’ perceived abilities for physical activity or exercise with the score ranging from 4 to 20 (mean = 15.6, SD = 2.8). The last component includes items #10 (“I have strong social skills”) and #11 (“I have strong communication skills”). Component #3 elaborates older adults’ sociality around physical activity or exercise with the score ranging from 2 to 10 (mean = 7.9, SD = 1.5). The total score ranged from 11 to 55, while the mean was 45.1 and the standard deviation was 6.1. The composite scores (total and component scores) are treated as in the continuous scale in the analysis.

The mean scores of the attitude component and sociality component in the SPPLI did not reach statistically significant differences by gender. However, the gender difference was close to the statistical significance in the physical activity ability component (*t* = 1.95, df = 339, *p* = 0.051). When examining physical literacy differences by education attainment, the participants who completed high school or college (including graduate degrees) had significantly higher physical activity ability scores than those who did not complete high school education (*t* = 4.20, df = 213, *p* < 0.001; *t* = 2.79, df = 228, *p* < 0.01, respectively) (Table 1). The results show that older adults with more years of education have better ability to perform physical activity, as is consistent with the literature [19]. The Student’s *t*-test results showed that participants who were in the maintenance stage had a significantly higher total score (*t* = 4.16, df = 339, *p* < 0.001) in each of the component scores (*t* = 3.68, df = 339, *p* < 0.001; *t* = 3.46, df = 153, *p* = 0.001; *t* = 2.35, df = 153, *p* < 0.05, respectively) than their counterparts. The results are consistent with the findings from Holler and associates’ 2019 study.

## 4. Discussion

Applying an appropriate method to measure physical competence and vitality under the concept of physical literacy for older adults aligns with the goal of healthy aging. Hence, it is important for researchers and practitioners to review the definition and adopt a measurement that differs from that used for youth or adults [20]. This study validated the Senior Perceived Physical Literacy Instrument (SPPLI), refined from the Perceived Physical Literacy Instrument (PPLI) initially developed for adults [16]. The results showed that the SPPLI has good reliability and validity to assess seniors’ physical literacy, using three components, i.e., attitude toward physical activity, physical activity ability, and sociality around physical activity.

The results showed that SPPLI has three components. Though the names of the three components are not the same as the ones used in previous studies, they are similar. For example, Roetert and Ortega [11] claimed motivation, confidence and knowledge as the three dimensions of physical literacy in older adults. Attitude and motivation are highly comparable. Since both are non-intelligence factors, they could affect each other [21,22,23]. In addition, a study has shown a positive relationship among attitude, moderate confidence, intention, and belief [24]. That is, having higher confidence or self-efficacy might lead to more positive attitudes toward exercise or physical activity [25]. Attitude and behavior have been defined as interdependent, since attitude is the output of interaction between personality and values. Attitude would also prevent people from exposure to negative information or events, leading to greater egoistic behavior [26]. Attitude can also provide important interventions to increase levels of physical activity, such as with the frequency of participation and self-efficacy [24].

Knowledge leads to the ability to acquire, transform, and apply information [27]. We believe that older adults’ physical activity ability is established based on their cumulative knowledge from their life experiences and education. Their ability could depend upon the process of knowledge acquisition and application [28]. Physically literate individuals could transfer knowledge to different conditions, especially when solving previously unencountered problems [29]. A study shows the importance of frequency, types, and duration of physical activity assessment for maintaining healthy behaviors based on the knowledge of the physical activity [30].

Besides attitude and ability, sociality is another component of the SPPLI. While attitude and ability are an individual’s characteristics, sociality involves interactions with others, and it plays a role in older adults’ physical literacy. During exercise, they might need to communicate with peers, know how to ask teachers questions, or even teach others [31]. Seeking social interactions may also be a factor for motivating older adults to exercise. Hence, social interaction becomes crucial for the elderly. The results are consistent with other studies. Keegan et al. [8] showed that social capability is one of the elements of physical literacy. They, thus, recommend including selected, specific sociality that can support the construct of physical literacy for older adults. A physically literate person needs to have social capability to promote health and physical activity throughout their lifespan. Campelo and Katz [12] also showed that the social domain of physical literacy includes family interaction, peer interaction, and therapist/staff interaction, etc. Hence, having group exercises or adding social interactions to exercise programs might increase older adults’ interest in participating [32]. In Taiwan, the policy of the Twelve-year Core Literacy Directed Physical Education Curriculum is aimed at promoting students’ physical literacy during senior high school, including “social participation” as one of the principles [33]. It can be observed that social participation in Taiwan has been cultivated since childhood, and the elderly should not be ignored. The results of this study also support this idea.

As for the gender issue, there is no difference in the SPPLI total score and its three component scores. The study showed that there is no difference in physical activity participation between male and female older adults, even though they both have positively perceived the advantages of physical activity [34]. For female participants, a study found that those women over 65 years old were more likely to exercise regularly and had more leisure time to exercise after retirement [35]. Women are more likely to engage in lower-intensity and less physical activity in the community [36]. Therefore, we recruited more female participants than males in this study. In contrast, at the same age and with similar levels of physical activity, the physical ability of men is usually higher because of physiological differences [37]. However, the assessment of physical fitness was not included in this study. Therefore, gender is not a critical factor that affects the physical literacy of seniors, since the physical literacy of seniors concerns one’s self-awareness rather than level of performance.

## 5. Limitations

There are some limitations of this study. First, most of the participants had completed junior high school or above and were literate. This study did not intend to compare the difference between literate and non-literate older adults. Second, because the participants were mostly recruited from community exercise classes, participants had some degree of exercise habits. Participants in this study could be more inclined to exercise or perform physical activity. Third, the number of female participants was larger than males. Because women tend to share their emotions more openly with others and have greater sociality [38], having a larger proportion of women in the sample might affect the sociality scores. These limitations may be improved by the expansion of the sampling pool, and the use of random sampling and/or cluster sampling in future studies.

## 6. Conclusions

As the population of older adults increases, the issue of aging cannot be ignored. Before improving the physical activity level of seniors, attention to physical literacy is critical, since it has been proven to be related to physical activity. This study developed the Senior Perceived Physical Literacy Instrument (SPPLI) based on the 18-question Perceived Physical Literacy Instrument (PPLI). Through PCA, researchers revised the original 18-question instrument to one with 11 questions. The internal consistency reliability and content and construct validity assessments demonstrated that the SPPLI possesses the proper reliability and validity to assess physical literacy among older adults. Therefore, this instrument is suggested for the physical literacy assessment of seniors.

It was found, by conducting an EFA, that the questions of the SPPLI contain three key elements: attitude, ability and sociality. Thus, the instrument could also be used in physical activity programs to assess the needs of older adults and/or the effectiveness of an intervention program that aims to improve the attitude, ability, and sociality of physical activity. However, whether it is effective to link physical literacy and physical activity programs still requires verification. Hence, future studies might further explore the relationship between physical literacy and physical activity in older adults.

## Figures and Tables

**Table 1 ijerph-19-06726-t001:** Characteristics of participants (*n* = 341).

Demographic			*n*	(%)	Total Score	Attitude Score	Ability Score	Sociality Score
					Mean ± SD	Mean ± SD	Mean ± SD	Mean ± SD
	**Age (years),**	74 ± 7.5						
Gender	Male	78 ± 8.1	77	22	45.4 ± 5.6	21.7 ± 2.7	15.7 ± 2.8	7.9 ± 1.3
	Female	93 ± 7.4	264	78	45.0 ± 6.2	21.5 ± 2.9	15.5 ± 2.9	7.9 ± 1.5
Education	Below senior high school		122	36	44.3 ± 6.7	21.7 ± 2.9	14.7 ± 2.9	7.7 ± 1.6
	Senior high school		95	28	45.6 ± 5.1	21.3 ± 2.3	16.3 ^a^ ± 2.2	7.8 ± 1.4
	University or above		124	36	45.5 ± 6.1	21.6 ± 2.9	15.8 ^b^ ± 2.5	8.1 ± 1.3
Exercise stages	Maintenance		244	72	46.0 ^c^ ± 6.9	22.0 ^c^ ± 3.3	16.0 ^c^ ± 3.1	8.0 ^c^ ± 1.6
	Non-maintenance		97	28	43.0 ± 5.4	20.7 ± 2.6	14.7 ± 2.6	7.6 ± 1.4

^a^ Participants with a high school diploma had a significantly higher mean score than those with an education below high school (*p* < 0.001). ^b^ Participants with college or graduate degrees had a significantly higher mean score than those with an education below high school (*p* < 0.01). ^c^ Participants in the maintenance stage had a significantly higher mean score than those in the non-maintenance stage (*p* < 0.001).

**Table 2 ijerph-19-06726-t002:** Initial factor structures by exploratory factor analysis (*n* = 238).

	Items	Component 1	Component 2	Component 3	Alpha
Q15	I can turn doing sports into an on-going habit of life	0.837	0.233	0.117	0.907
Q14	I have a mindset for lifelong sports	0.830	0.219	0.256	
Q9	I am willing to do sports for better health	0.821	0.251	0.140	
Q17	I am aware of the benefits of sports related to health	0.730	0.240	0.244	
Q16	I establish friendship through sports	0.726	0.204	0.291	
Q4	I have a positive attitude and interest in sports	0.610	0.490	0.196	
Q5	I appreciate myself or others doing sports	0.565	0.419	0.158	
Q18	I aspire to know the current sports trend	0.397	0.315	0.349	
Variance explained 28.56%					
Q2	I am physically fit, in accordance to my age	0.119	0.799	0.218	0.890
Q1	I possess adequate fundamental movement skills	0.292	0.766		
Q6	I am able to apply PE knowledge in the long run	0.284	0.726	0.164	
Q3	I am able to apply learnt motor skills to other physical activities	0.271	0.703	0.199	
Q8	I possess self-evaluation skills for health	0.251	0.650	0.494	
Q7	I possess self-management skills for fitness	0.388	0.638	0.347	
Variance explained 24.64%					
Q11	I have strong social skills	0.241	0.214	0.837	0.801
Q10	I have strong communication skills	0.266	0.209	0.826	
Variance explained 14.30%					

Total variance explained: 67.51%. Note: the Cronbach’s alpha for this scale was 0.93.

**Table 3 ijerph-19-06726-t003:** Second factor structures by exploratory factor analysis (*n* = 238).

Factor Name	Original Item	New Item	Items	Component 1	Component 2	Component 3	Alpha
Attitude	Q15	SQ1	I can turn doing sports into an on-going habit of life	0.855	0.244	0.110	0.909
	Q14	SQ2	I have a mindset for lifelong sports	0.847	0.212	0.251	
	Q90	SQ3	I am willing to do sports for better health	0.823	0.266	0.133	
	Q17	SQ4	I am aware of the benefits of sports related to health	0.753	0.238	0.219	
	Q16	SQ5	I establish friendship through sports	0.721	0.222	0.277	
Variance explained 32.60%							
Ability	Q1	SQ6	I possess adequate fundamental movement skills	0.298	0.780		
	Q2	SQ7	I am physically fit, in accordance to my age	0.131	0.773	0.225	0.834
	Q6	SQ8	I am able to apply PE knowledge in the long run physical activities	0.245	0.765	0.176	
	Q3	SQ9	I am able to apply learnt motor skills to other	0.271	0.745	0.214	
Variance explained 24.70%							
Sociality	Q11	SQ10	I have strong social skills	0.241	0.218	0.848	0.801
	Q10	SQ11	I have strong communication skills	0.275	0.209	0.841	
Variance explained 16.11%							
Total variance explained 73.43%							

Note: the Cronbach’s alpha for this scale was 0.90.

**Table 4 ijerph-19-06726-t004:** Factor structures by exploratory factor analysis (*n* = 341).

Factor Name	New Item	Items	Component 1	Component 2	Component 3	Alpha
Attitude	SQ1	I can turn doing sports into an on-going habit of life	0.856	0.215	0.137	0.911
	SQ2	I have a mindset for lifelong sports	0.829	0.190	0.261	
	SQ3	I am willing to do sports for better health	0.796	0.270	0.109	
	SQ4	I am aware of the benefits of sports related to health	0.765	0.224	0.181	
	SQ5	I establish friendship through sports	0.720	0.221	0.309	
Variance explained 31.8%						
Ability	SQ6	I possess adequate fundamental movement skills	0.292	0.790		0.839
	SQ7	I am physically fit, in accordance to my age	0.239	0.770	0.194	
	SQ8	I am able to apply exercise knowledge in the long run physical activities	0.143	0.756	0.239	
	SQ9	I am able to apply learnt motor skills to other	0.214	0.755	0.199	
Variance explained 24.5%						
Sociality	SQ10	I have strong social skills	0.243	0.220	0.849	0.809
	SQ11	I have strong communication skills	0.284	0.215	0.835	
Variance explained 16.1%						
Total variance explained 72.5%						

Note: the Cronbach’s alpha for this scale was 0.90.

**Table 5 ijerph-19-06726-t005:** Senior Perceived Physical Literacy Instrument and the item response distributions.

Items		Mean Score	Standard Deviation
1.	I can turn doing sports into an on-going habit of life	4.3	0.7
2.	I have a mindset for lifelong sports	4.3	0.7
3.	I am willing to do sports for better health	4.4	0.7
4.	I am aware of the benefits of sports related to health	4.5	0.6
5.	I establish friendship through sports	4.2	0.8
6.	I possess adequate fundamental movement skills	3.9	0.9
7.	I am physically fit, in accordance to my age	3.8	0.9
8.	I am able to apply PE knowledge in the long run	4.0	0.9
9.	I am able to apply learnt motor skills to other physical activities	4.0	0.8
10.	I have strong social skills	3.9	0.9
11.	I have strong communication skills	4.1	0.8

Note: the Cronbach’s alpha for this scale was 0.90.

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
