# Peer review of "Psychometric Validation of Senior Perceived Physical Literacy Instrument"

_ijerph, 2022, doi:10.3390/ijerph19116726_

Round 1

Reviewer 1 Report

I would just like the authors to adjust the tables so that they don't break throughout the text.

Author Response

We appreciate that Reviewer #1 recommend. We have tried our best to adjust the table space and not to break the table. Please refer to the text again. (including Line 41-45,66-72, 95-97, and table 2- 4)

Reviewer 2 Report

Dear Authors,

The current revised version of the text is in my opinion more Readers-friendly than the previous one. Although several reviewers expressed a series of improvement suggestions, the Authors dismissed many of them and did not put much effort in improving their text. They just made the location of the sub-sections more logical, improved coherent the use of abbreviations through the text and the labeling of the Tables, added more details to the Methodology and removed repetitive text fragments - while leaving the rest untouched, including the list of references. However, the final positive effect exceeds the expectations based on the limited means applied by the Authors in order to achieve it.

Author Response

We thank Reviewer #2 for the comment. We had try to increase some descriptions to make the text can be more clear and more logical (including Line 41-45,66-72, 95-97, table 3, table 4 and conclusions). Please refer to the text again.

Reviewer 3 Report

I would like to thank the authors for the changes in the last version. Nevertheless, I highlight some comments indicated in the first revision.

Theoretical framework:

In the first revision it was indicated:

Revision 1. The authors describe the concept of physical literacy on the basis of components associated with different dimensions according to the instrument used in groups of people of different ages. This section is not discussed, nor is it exhaustively developed in the theoretical framework.

The theoretical framework should go more deeply into the different dimensions and explain by means of an exhaustive review (mentioning a systematic review, meta-analysis or review of the existing literature in this field) the problem it proposes to analyse.

Revision 2. Although there is no systematic review on the topic of study, the authors should develop further the theoretical framework on the dimensions they are researching.

Objectives:

In the first revision, it was said: The last paragraph of the theoretical framework should specify the object of study and perhaps some hypotheses.

Revision 2. This section is not described in the last paragraph of the theoretical framework of the new version.

Methodology

Revision 2. The authors have partially responded to the requests indicated in the first revision.

Results

In the first revision it was indicated: 

Lines 163-166

Very risky statement. According to the questionnaire used, it can only be stated that the respondents have a perception of their capacity for physical activity and not a real capacity for physical activity.

Revision 2.  It should be reworded, without the statement being so risky, on the basis of the gathered evidence.

Conclusions

Revision 2. This section remains unchanged. Authors should try to improve it. 

Author Response

Reviewer #3

Although there is no systematic review on the topic of study, the authors should develop further the theoretical framework on the dimensions they are researching.

We thank Reviewer #3 for the comment. We add a literature review description on Line 66-72 to develop a further theoretical framework of the text.

The last paragraph of the theoretical framework should specify the object of study and perhaps some hypotheses.

We thank Reviewer #3 for the comment. We add the object and hypotheses of study in the text.(Line 71-72)

The authors have partially responded to the requests indicated in the first revision.

We thank Reviewer #3 for the comment. We try hard to revise the article to the most perfect.

Line 163-166. Very risky statement. According to the questionnaire used, it can only be stated that the respondents have a perception of their capacity for physical activity and not a real capacity for physical activity.

We thank Reviewer #3 for the comment. We change the sentence as “The component #2 illustrates older adult’s perceived abilities toward physical activity or exercise with the score ranged from 4 to 20 (mean=15.6, SD=2.8).” .(Line 176-178) To illustrate the self-cognition capacity of physical capacity.

Although this was a subjective measure, just like "The Borg Rating of Perceived Exertion (RPE)", which was a way of measuring physical activity intensity level and was used widely (eg. US CDC ).

https://www.cdc.gov/physicalactivity/basics/measuring/exertion.htm

Conclusions

This section remains unchanged. Authors should try to improve it. 

We thank Reviewer #3 for the comment. We made some revisions to make the article to the most perfect, please refer to the text again. .(Line 251-266)

Round 2

Reviewer 3 Report

This article deals with a subject of great importance, which is to study the situation of physical activity in the elderly. This group of people has been growing progressively in recent years and will continue to do so in the coming decades.

In the past version, the study presented several weaknesses that hinder a positive assessment. Now in the last version, some issues have been improved but not all the remarks have been answered and included.

Theoretical framework:

Observations past review:

The authors describe the concept of physical literacy on the basis of components associated with different dimensions according to the instrument used in groups of people of different ages. This section is not discussed, nor is it exhaustively developed in the theoretical framework.

The theoretical framework should go more deeply into the different dimensions and explain by means of an exhaustive review (mentioning a systematic review, meta-analysis or review of the existing literature in this field) the problem it proposes to analyse.

The theoretical framework also fails to develop any section referring to characteristics of the population studied, such as whether there is a relationship between the type of academic background and the practice of physical activity; such as the difference between men and women in the performance of a regular physical exercise.

Observations new review

This section has been improved, not excellently but it is acceptable as it is in this latest version.

Objectives:

Observations past review:

The last paragraph of the theoretical framework should specify the object of study and perhaps some hypotheses.

Studying only people's perception of their physical literacy is too poor a topic for an article of this kind to be published in a journal of high scientific impact.

It would have been very interesting to complete this study by asking about the past and current physical activity habits of these people. Type of physical activity, number of times per week they do it and conditions of practice. Certainly, the perception of the relationship with others is not the same for people who practise an individual physical activity as it is for those who practise a collective activity.

 Observations new review

This section has been improved. After the objective, the hypothesis could have been written in hypothesis format, in a more explicit way.

Methodology

Observations past review:

 There are different problems in organising the information.

The section on ethical considerations, which is described in the first paragraph, is best placed at the end of the section on participants or in a specific section on ethical considerations.

Authors should separate the section on participants and procedures.

 Observations new review

This point has not been answered

Participants

Observations past review:

Participants should include the mean, standard deviation, as well as age range. They should also consider the percentage of male and female respondents.

 Observations new review

This point has not been answered

Include a section on Instruments and Procedures.

Observations past review:

Explain better the conditions of the research team that passed the questionnaire, the number of people, and experience in this field.

 Observations new review

This point has not been answered

Data analysis

The authors should justify why they did not do other psychometric tests, such as using another form to assess the convergent validity of the questionnaire.

Results

Observations past review:

This information shown in the results on lines 108-110 should go in the participant's section.

Lines 120-122

This data should go in the methodology section.

 Observations new review

This point has not been answered

Observations past review:

Lines 163-166

Very risky statement. According to the questionnaire used, it can only be stated that the respondents have a perception of their capacity for physical activity and not a real capacity for physical activity.

Observations new review

This point has  been answered

Discussion (the title should be written in the singular)

Observations new review

The authors should start the discussion by recalling the aim of the research.

The discussion does not mention anything about the gender perspective in this study.

In lines 189-191 the knowledge dimension is poorly developed in the discussion.

In general, the discussion is very poor and does not go into the problems of the study in depth.

 Observations new review

This point has not been answered

Conclusions

Observations past review:

This section is also very poor.

 Observations new review

This section has been improved, not excellently but it is acceptable as it is in this latest version.

Author Response

Thanks the editor and reviewer #3  for spending the time reviewing our manuscript. We replied the questions and suggestion in the following.

No.

Reviewer comment

Author response

Reviewer #3

1.

Theoretical framework

This section has been improved, not excellently but it is acceptable as it is in this latest version.

We thank Reviewer #3 for affirmative.

2.

Objectives

 This section has been improved. After the objective, the hypothesis could have been written in hypothesis format, in a more explicit way.

We thank Reviewer #3 for the comment. We write the hypothesis in hypothesis format in line 69-72. Please refer to the text again.

3.

Methodology

The section on ethical considerations, which is described in the first paragraph, is best placed at the end of the section on participants or in a specific section on ethical considerations.

Authors should separate the section on participants and procedures.

We thank Reviewer #3 for the comment. We adjust the paragraph of the section and separate the section on participants and procedures. Please refer to the text again.( line 78-80)

4.

Participants

Participants should include the mean, standard deviation, as well as age range. They should also consider the percentage of male and female respondents.

We thank Reviewer #3 for the comment. We add the mean, standard deviation in table 1. Please refer to the text again. Besides, the average scores of males and females are not significantly different, so the researcher believes that gender does not affect the results. Therefore, the scale does not distinguish between genders, so this study  adopted the statistical analysis of mixed male and female.

5.

Include a section on Instruments and Procedures.

Explain better the conditions of the research team that passed the questionnaire, the number of people, and experience in this field.

We thank Reviewer #3 for the comment. We explain he conditions of the research team that passed the questionnaire, the number of people, and experience in this field in line 102-107. Please refer to the text again.

6.

Data analysis

The authors should justify why they did not do other psychometric tests, such as using another form to assess the convergent validity of the questionnaire.

We thank Reviewer #3 for the comment. We thank Reviewer #3 for the comment. We had tried the other form (Confirmatory Factor Analysis, CFA) to assess the convergent validity of the questionnaire. The graph path presented by the CFA analysis is as follows. Each factor has a high correlation with its relative items, and there is no significant correlation between the factors. The results of the CFA analysis are consistent with the original results of the EFA analysis , so we decided to keep the original result. We explained the reasons why we used this psychometric test on line 58-61.

7.

This information shown in the results on lines 108-110 should go in the participant's section.

We thank Reviewer #3 for the comment. We change lines 108-110 to the participant's section. Please refer to the text again.

8.

Lines 120-122

This data should go in the methodology section.

We thank Reviewer #3 for the comment. We change lines 120-122 to the methodology section. Please refer to the text again.

9.

Lines 163-166

This point has  been answered

We thank Reviewer #3 for affirmative.

10.

The authors should start the discussion by recalling the aim of the research.

We thank Reviewer #3 for the comment. We recall the aim of the research at the start of the discussion in lines 214-219.

11.

The discussion does not mention anything about the gender perspective in this study.

We thank Reviewer #3 for the comment. Because there is no difference in SPPLI total score and its three component scores. We added the gender section at discussion in lines 265-277. Please refer to the text again.

12.

In lines 189-191 the knowledge dimension is poorly developed in the discussion.

We thank Reviewer #3 for the comment. We add more description of the knowledge dimension in lines 241-245.

13.

In general, the discussion is very poor and does not go into the problems of the study in depth.

We thank Reviewer #3 for the comment. Beside added more description of the attitude dimension in lines 228-236, we add deeper description of the ability and sociality in lines 234-240 and 260-264 separately.

14.

Conclusions

This section has been improved, not excellently but it is acceptable as it is in this latest version.

We thank Reviewer #3 for affirmative.

This manuscript is a resubmission of an earlier submission. The following is a list of the peer review reports and author responses from that submission.

Round 1

Reviewer 1 Report

Despite the importance of validating an instrument of this nature, which will seek to assess the health literacy of a specific population, I leave here some important comments to improve this manuscript:

  1. The introduction needs substantial improvements. There are some sentences that have a similar meaning, sentences should be checked in relation to English language, and more articles on scale validation studies that assess the similar domain should be cited.

  1. The paragraph between lines 46 to 55 cites a validated scale for children (?). I recommend that you remove it from the article and focus on cited studies with validated scales for the elderly population.

  1. A study that presents an exploratory analysis cannot be considered a validation study. You have to run another specific analysis (Structural Equation Model and /or confirmatory factor analysis) so that you can confirm the process. I recommend that you publish everything in the same and do not share results.

  1. In the table 1, will be necessary to review the scores of scales’ subdimension because the standard deviation is not included.

  1. Authors have to create session and tell a story about this instrument. Who is the original author? How was it created? What does it measure? What are the values (range) of for each dimension? Does it have a total score? Can it be analyzed as a continuous or categorical scale of values in population studies? What was the Cronbach alpha value of other scales that have been validated? In which countries is it used?

Reviewer 2 Report

Dear Authors,

Congratulations to develop such an important topic and to detect this gap in the literature. However, some I do have some concerns, that I exposed below.

Introduction

The introduction is clear and highlight the gap in the literature.

Methods and Materials

Participants: did the authors checked the if there would be no cognitive decline in the selected elderly?

The considered “pilot study” had a greater sample than the study validation? Did this sample include the same subjects (I presume not, considering line 85)?

It could be my perception, but I would suggest to re-write the paragraph between line 69 and 80 because it starts with one study, then emerges the pilot study, and again the present study

Line 71 – “they responded to PPLI or SPPLI” in the pilot study or in the present study? In fact, how many elderlies answered to the first and the second? Because in line 385 and in results it was assumed that all of them answered the SPPLI.

Results

Line 112 – could these results be influenced by the fact that those subjects are active?

Please format table 1

Table 1 – although it was mentioned a huge difference in que quantity of men and women in this sample, I would suggest presenting those results separately by sex.

Line 120 – it was already mentioned above what the acronym corresponds to, so I would remove the description. The same in line 122 to PCA. Line 125 and 131 too.

Line 123-24 – please present the result as in table 2 this value was not expressed. The same for line 126

Table 5 – please format the table and its title.

Limitation

I think that the limitations pointed out could really affect the results of this study, as literate people and active people could biased the results. To overcome this issue, perhaps it would be better to dub this study a "pilot study" to serve as motivation for a new, larger, and more balanced sample. 

Reviewer 3 Report

The manuscript consists of total 8 pages, including 5 tables and the 

list of total 30 references. The original material-based manuscript 

presents the results of the cross-sectional study aiming at Senior

Perceived Physical Literacy Instrument validation which fits into the 

scope of the topics raised in the Journal. The topic is important in the 

context of the ageing society phenomenon and it is good that more 

and more tools are available that serve the evaluation the factors 

defining the fitness of the elderly. 

It is highly unfortunate but I have to write that the article in its 

current form presents as a quite heavy read, mainly due to the

manner of writing chosen by the Authors: the actual ideas presented 

in the text are very difficult to follow and reading becomes quickly 

quite frustrating for Readers as the Authors packed most the 

sentences richly with names of authors and dates of the referred 

publications along with long inclusions of numerical statistics 

parameters - which in case of many of the text lines leaves pretty 

little space for actual communicates and blurs their sense. I assume 

that changing the style of referring to the literature to the [X] brackets 

based one and focusing in the narration more on the actual meaning 

of the results rather than citing them directly in the text (it may be 

better to put them in the tables with relevant comments) would help 

the Authors a lot in clarifying the text without taking away any of its 

unquestionable value. The text shall undergo a review and correction, 

optimally by a native English speaker, as there are many places in the 

text where the sentences are unclear or traces of word-for-word 

translations without proper relation of the chosen words to the topic 

context are distinguishable.

The Abstract is structured and it presents as an adequate mirror of 

the crucial ideas confined in the main text.

The Introduction section is short but difficult to follow for the Reader 

as the same statements that are repeated many times in the row 

without any visible justification. In the lines 33/34 there is a repeated 

sentence.

The Methods and Materials section contains detailed information on 

the Authors' proceedings.

The Results section is very difficult to follow for Readers as it does 

not guide them easily enough through what and in what sentence is 

presented, the general impression is of quite chaotic communication. 

Maybe if the Authors try out introducing some sort of punctuation in 

discussing each of the following aspects/groups of questions would 

help the text to make a more organized impression. The Authors use 

the names of the tools and their abbreviations interchangeably 

through the text - it is good and friendly practice to remind the

Reader of the meaning of the abbreviation while used for the first 

time and then on each new page but in the current version of the text 

it seems to be rather random practice, which does not help the clarity 

of the text.

The Discussions section inherit the style and format of writing used in 

previous sections and thus it also presents as a quite difficult read. 

The Conclusion section is based on the discussed results.

The captions of the Tables need to be improved as in Table 1 it is 

unclear what tool has been used to get the scores; in Table 2, 3 and 4 

it is unclear what the "Q" and "F" mean without browsing in the text 

and their sequence is irregular in the table.

The References need to be checked for the format coherence (e.c.

position 7), there are 31 positions but only 30 actual references as 

the position No 26 is typing mistake (the line numbered as an 

additional position is in fact the part of the reference No 25).

The Authors may want to consider broadening the background by 

referring to the following aspects and sources:

- the social and demographic background of the ability to maintain fitness in

the old age, as it is e.c. in: https://doi.org/10.3390/ijerph19042066 

- importance of the fitness in the old age to maintain the desired 

position in the ageing society, as it is e.c. in: 

https://doi.org/10.3390/su14052533 and 

https://doi.org/10.3390/socsci11020041 as well as an acceptable 

quality of life, as it is e.c. in: 

https://doi.org/10.3390/ijerph182211816

- the importance of organized and perpetuated exercise in keeping the

elderly active, both classic as it is e.c. in: 

https://doi.org/10.3390/ijerph19042178 and modern, as it is e.c. in: 

https://doi.org/10.3390/ijerph182412939

- other tools in the same research field, as it is e.c. in: 

https://doi.org/10.3390/ijerph19041937

Reviewer 4 Report

This article deals with a subject of great importance, which is to study the situation of physical activity in the elderly. This group of people has been growing progressively in recent years and will continue to do so in the coming decades.

The study presents several weaknesses that hinder a positive assessment.

Theoretical framework:

The authors describe the concept of physical literacy on the basis of components associated with different dimensions according to the instrument used in groups of people of different ages. This section is not discussed, nor is it exhaustively developed in the theoretical framework.

The theoretical framework should go more deeply into the different dimensions and explain by means of an exhaustive review (mentioning a systematic review, meta-analysis or review of the existing literature in this field) the problem it proposes to analyse.

The theoretical framework also fails to develop any section referring to characteristics of the population studied, such as whether there is a relationship between the type of academic background and the practice of physical activity; such as the difference between men and women in the performance of a regular physical exercise.

Objectives:

The last paragraph of the theoretical framework should specify the object of study and perhaps some hypotheses.

Studying only people's perception of their physical literacy is too poor a topic for an article of this kind to be published in a journal of high scientific impact.

It would have been very interesting to complete this study by asking about the past and current physical activity habits of these people. Type of physical activity, number of times per week they do it and conditions of practice. Certainly, the perception of the relationship with others is not the same for people who practise an individual physical activity as it is for those who practise a collective activity.

Methodology

 There are different problems in organising the information.

The section on ethical considerations, which is described in the first paragraph, is best placed at the end of the section on participants or in a specific section on ethical considerations.

Authors should separate the section on participants and procedures.

Participants

Participants should include the mean, standard deviation, as well as age range. They should also consider the percentage of male and female respondents.

Include a section on Instruments and Procedures.

Explain better the conditions of the research team that passed the questionnaire, the number of people, experience in this field.

Data analysis

The authors should justify why they did not do other psychometric tests, such as using another form to assess the convergent validity of the questionnaire.

Results

This information shown in the results on lines 108-110 should go in the participant's section.

Lines 120-122

This data should go in the methodology section.

Lines 163-166

Very risky statement. According to the questionnaire used, it can only be stated that the respondents have a perception of their capacity for physical activity and not a real capacity for physical activity.

Discussion (the title should be written in the singular)

The authors should start the discussion by recalling the aim of the research.

The discussion does not mention anything about the gender perspective in this study.

Lines 189-191 the knowledge dimension needs to be developed in the discussion.

In general, the discussion needs to be improved and does not go into the problems of the study in depth.

Conclusions

This section needs to be improved.